# CROSS-MODALITY DEBIASING: USING LANGUAGE TO MITIGATE SUB-POPULATION SHIFTS IN IMAGING

## ABSTRACT

Sub-population shift is a specific type of domain shift that highlights changes in data distribution within specific sub-groups or populations between training and testing. Sub-population shift accounts for a significant source of algorithmic bias and calls for distributional robustness. Recent studies found inherent distributional robustness in multi-modality foundation models, such as the vision-language model CLIP, yet this robustness is vulnerable through parameter fine-tuning. In this paper, we propose leveraging the connection of robustness among different modalities and reshaping the distributional robustness of one modality with another. Specifically, in the context of the distributional robustness of CLIP, we propose to leverage natural language inputs to debias the image feature representations, to improve worst-case performance on sub-populations. Our extensive empirical studies show that image representations debiased by natural language can achieve significant performance improvement and reduction of performance instability under sub-population shifts.

## 1 INTRODUCTION

The domain shift between the training data and data at the inference stage is commonly found in machine learning systems. For instance, in applications such as robotics James et al. (2019); Wulfmeier et al. (2017), navigation Lütjens et al. (2019); Bharadhwaj et al. (2019), and auto collecting Moreno-Torres et al. (2012), training data may be collected from simulated or misaligned environments to reduce cost. The collected data will likely encounter *domain shift* during deployment, where the model robustness towards those shifts is a crucial requisite for safety deployment. *Sub-population shift* is a specific type of domain shift where there are changes in the distribution of data within specific sub-populations or groups Koh et al. (2021); Santurkar et al. (2020). Sub-population shift leads to generalization issues of groups and constitutes a significant source of algorithmic bias. A robust model against sub-population shift shall perform well on various demographic groups, such as groups identified by gender, race, or age, no matter the group population. However, classical methods such as empirical risk minimization are known to be fragile under such sub-population shifts or domain shifts Duchi & Namkoong (2018); Rockafellar et al. (2000).

More recently, the sub-population shift has also been identified as a critical issue in large foundation models Henderson et al. (2023); Gan et al. (2022). One notable example of cross-modality foundation models is Contrastive Language-Image Pre-training (CLIP) Radford et al. (2021). CLIP is a neural network trained on pairs of images and natural language supervision. It enables learning rich feature representations for images and leverages natural language cues during training and inference. Consequently, methods built upon CLIP have gained considerable attention due to their ability to align diverse modalities and address various tasks. However, recent studies found that fine-tuning CLIP improved task performance but often compromised its original robustness to domain shifts Wortsman et al. (2022); Kumar et al. (2022); Zhang et al. (2021). Similarly, the same phenomenon regarding robustness to sub-population shifts was observed Li et al. (2021); Zhang & Ré (2022); Lee et al. (2022). These findings collectively indicate that the careful construction of feature embeddings can be easily compromised by the negligent utilization of label information during training and emphasize the importance of investigating suitable methods for handling distributional shifts in conjunction with multimodal models like CLIP.

Distributionally Robust Optimization (DRO) provides a paradigm to tackle the naturally distributional shifts Namkoong & Duchi (2016); Rockafellar et al. (2000). DRO typically involves optimizing

the model performance by considering the *uncertainty* of data, which approximates the worst-case shift in domain features or sub-populations from the training distribution. The shift is usually bounded within a distributional divergence distance, such as $f$-divergences Ben-Tal et al. (2013); Namkoong & Duchi (2016); Hashimoto et al. (2018); Duchi & Namkoong (2018); Shapiro (2021) and Wasserstein distances Gao et al. (2017); Blanchet et al. (2019); Kuhn et al. (2019); Sinha et al. (2017). However, a straightforward combination of CLIP with universal DRO approaches, such as $\chi^2$-DRO Hashimoto et al. (2018) and its variants, may introduce inherent risks like unstable performance and the requirement for a domain-aware validation dataset. Therefore, it is essential to explore alternative strategies to mitigate distributional shifts while preserving the effectiveness and robustness of CLIP.

A recent study by Dunlap et al. (2022) introduced an approach that modifies the feature embedding of classifiers utilizing CLIP as a backbone. This modification enables the extension of classifier capabilities to previously unseen domains by leveraging natural language descriptions associated with these unseen domains. The example demonstrated the potential of adjusting distributional robustness using interconnected modalities. In this paper, we study strategies for mitigating sub-population shifts by leveraging natural language supervision in the context of the language-image foundation model CLIP. Notably, our investigation is conducted under the assumption of a domain-oblivious setting, wherein the sub-population membership of individual instances remains unknown during the training phase.

Our contributions are summarized as follows:

- We build a principled connection between natural language supervision and robustness to sub-population shift (also known as subgroup robustness) and provide extensive experimental analysis, which shows the capability of mitigating robustness issues in one modality by identifying and analyzing it in another modality.
- We show that without instance-wise label information, the proposed method consistently improves worst-case performance under sub-population shifts over original zero-shot learning of CLIP under divergent settings.

## 2  RELATED WORK

Recent years witnessed a growing interest in studying the distributional robustness of vision-language foundation models Fang et al. (2022); Nguyen et al. (2022); Gan et al. (2022). There is extensive evidence of robustness deterioration when applying classical fine-tuning methods, such as linear-probe, to pre-trained models, and there are great efforts to alleviate these robustness issues Wortsman et al. (2022); Gao et al. (2021); Kumar et al. (2022). To enhance performance in the presence of sub-population shifts, broadly used strategies that are based on loss values have been adapted to foundation models fine-tuning Zhang & Ré (2022). Moreover, there is a growing trend that natural language supervision is adapted in the training phase to debias learned feature representations Ranasinghe et al. (2022); Petryk et al. (2022); Wang et al. (2022). A recent work that aligns with our approach is presented in Zhang et al. (2023), which utilizes language to control model behaviors by identifying misclassified instances and influential attributes in the form of language descriptions and then continuing fine-tuning the vision classifier upon that information.

Distributionally robust optimization has been widely studied to handle the situation where the test distribution is undetermined Shapiro (2021); Namkoong & Duchi (2016); Quiñonero-Candela et al. (2008). Common approaches involve constructing an uncertainty set around the training distribution with some divergence to approximate the unknown distribution. Certain real-world scenarios, such as sub-population shift Koh et al. (2021); Santurkar et al. (2020); Jeong & Namkoong (2020), can be modeled as minimizing the supremum of the loss within the uncertainty set. Some convenient dual reformulations of the optimization problem in terms of some specific divergence are introduced such as $\chi^2$-DRO Hashimoto et al. (2018). In the context of addressing sub-population shifts, exploring statistical features to identify minority groups during training, through the analysis of gradients, losses, and feature spaces, has also gained popularity Liu et al. (2021); Nam et al. (2020); Sagawa et al. (2019); Sohoni et al. (2020).

## 3    BACKGROUND

Considering a machine learning task with given training distribution $P$ over input space $\mathcal{X} \in \mathbb{R}^d$ and vectorized label space $\mathcal{Y} \in \mathbb{R}^c$, empirical risk minimization aims to optimize the average performance of a model $f_\theta : \mathcal{X} \to \mathcal{Y}$ parameterized with $\theta$ over observed i.i.d. samples $\mathbf{z}_k = (\mathbf{x}_k, \mathbf{y}_k) \sim P$ for $k = \{1, ..., m\}$, which is formulated as $\min_\theta \mathbb{E}_{Z \sim P}[\ell(\theta, Z)]$. The problem of **sub-population shift** concerns the worst-case performance over some pre-defined sub-populations or domains $\{Q_1, ..., Q_n\}$ from an uncertainty set of distributions $\mathcal{Q} \in \mathcal{X} \times \mathcal{Y}$ where $P$ and $Q$ have the same support, and the objective is formulated as $\min_\theta \sup_{Q \in \mathcal{Q}} \mathbb{E}_{Z \sim Q}[\ell(\theta, Z)]$. In this paper, we use the vision-language foundation model CLIP Radford et al. (2021) in our evaluations and the proposed strategy and principle can be easily extended to other multimodal foundation models. We analyze the challenges associated with mitigating sub-population shift using $\chi^2$-DRO Hashimoto et al. (2018).

### 3.1    CLIP

Because of the rich image feature representations and its ability to incorporate natural language supervision during training and inference, CLIP has been widely studied and applied in various domains. In particular, the general zero-shot performance of CLIP serves as an essential baseline for algorithm development nowadays. The zero-shot classifier, which relies on CLIP as its backbone, utilizes the image encoder of CLIP and a linear classifier constructed using domain-specific descriptions for a given task. We formally define this zero-shot classifier as follows.

We denote $I_{\theta_i} : \mathcal{X} \to \mathcal{I}$ as the image encoder of CLIP that maps input space $\mathcal{X}$ to image embedding space $\mathcal{I}$, and $T_{\theta_t}(t) \in \mathcal{T}$ as the text encoder of CLIP that maps some text $t$ to text embedding space $\mathcal{T}$, where $\mathcal{I} \in \mathbb{R}^e$ and $\mathcal{T} \in \mathbb{R}^e$. We abbreviate $I_{\theta_i}(\cdot)$ and $T_{\theta_t}(\cdot)$ as $I(\cdot)$ and $T(\cdot)$ respectively. A set of classification text prompts, $\{t_1, ..., t_c\}$, that is empirically derived from class labels and class-domain description is necessary to construct a zero-shot classifier, e.g., $\{t_1, t_2\} = \{$"*a photo of a blond hair people*", "*a photo of a not blond hair people*"$\}$ for the common dataset CelebA Liu et al. (2015), we abbreviate $\{t_1, t_2\}$ as "*a photo of a* {not blond, blond} *hair people*". The zero-shot classifier built upon CLIP classifies an input $\mathbf{x}$ through: $\arg \max_{i \in [c]} I(\mathbf{x})^T T(t_i)$, where $[c]$ refers to the set of classes $\{1, 2, ..., c\}$. Various fine-tuning methods have been proposed to better adapt CLIP to downstream tasks, we select architecture design by Gao et al. (2021) for general training purposes. It added an extra feature adapter, $A_{\theta_a} : \mathcal{I} \to \mathcal{I}$, between $I(\cdot)$ and $T(\cdot)$, and we abbreviate $A_{\theta_a}$ as $A(\cdot)$. The corresponding training and inference are formulated as:

$$\textbf{Training} \quad \min_{\theta_a} \mathbb{E}_{(\mathbf{x}, \mathbf{y}) \sim P}[\ell_{\text{ce}}([(A_{\theta_a} \circ I(\mathbf{x}))^T T(t_i)]_{i \in [c]}, \mathbf{y})], \tag{1}$$

$$\textbf{Inference} \quad \arg \max_{i \in [c]} (A \circ I(\mathbf{x}))^T T(t_i), \tag{2}$$

where $[(A_{\theta_a} \circ I(\mathbf{x}))^T T(t_i)]_{i \in [c]}$ denotes a vector with $i$-th value as $(A_{\theta_a} \circ I(\mathbf{x}))^T T(t_i)$, and $\ell_{\text{ce}(\cdot)}$ denotes the cross-entropy loss. We abbreviate the network architecture as $I \triangleright A \triangleright T$ following the input stream from the image encoder of CLIP to the feature adapter and then to the text embedding.

### 3.2    DRO

Given a training distribution $P$ and a predefined divergence $D$, DRO aims to minimize the expected risk over the distribution $Q$ that is in a ball around the training distribution $P$ w.r.t. divergence $D$. The expected DRO risk is defined as $\mathcal{R}_{D;\rho}(\theta; P) = \sup_{\mathcal{Q} \ll \mathcal{P}} \{\mathbb{E}_{Z \sim Q}[\ell(\theta, Z)] : D(Q||P) \leq \rho\}$ for some $\rho > 0$ where $\mathcal{Q} \ll \mathcal{P}$ denotes $\mathcal{Q}$ is absolutely continuous with respect to $\mathcal{P}$. Supported by the convenient dual characterization of Cressie-Read family of Rényi divergence Shapiro (2017); Cressie & Read (1984), pioneering work eliminates the untouchable distribution $Q$ in $\mathcal{R}_{D;\rho}(\theta; P)$ and only exploits training distribution $P$ to solve the problem Duchi & Namkoong (2018); Zhai et al. (2021). Referring Zhai et al. (2021), we can show that, minimizing the loss values that are larger than a specific threshold, results in minimizing the DRO risk where $P$ and $Q$ have the same support, i.e., sub-population shift. This reinforces that the loss values (wrongly classified instances), based on over-sampling or re-weighting methods, are being used for mitigating the sub-population shift.

However, it is hard to directly apply DRO risk in the general training pipeline of deep models. Firstly, (Mini-batch) SGD is a biased estimation of DRO risk Ghosh et al. (2018), which suggests a two-phase training pipeline. Secondly, directly minimizing DRO risk is equal to minimizing the loss over a small

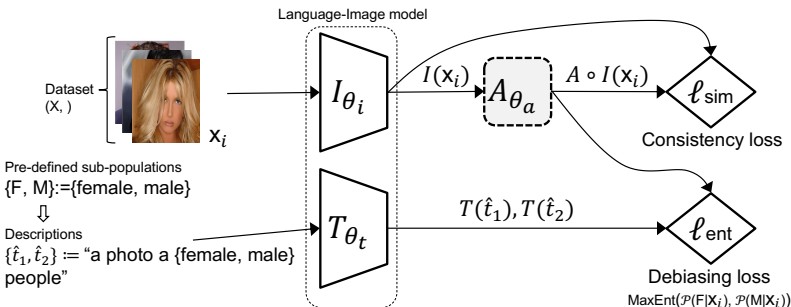

Figure 1: Training phase of L-DRO on CelebA Liu et al. (2015). Given the training dataset (without instance-wise label information) and concerned sup-populations, L-LDR aims to learn a feature adapter $A_{\theta_a}$ to transform image representation from original embedding to a debiased embedding. The goal is to ensure that the debiased embedding does not reveal any information about sub-population membership while minimizing significant changes from the original embedding.

portion of whole data points, which likely deteriorates the average performance and even worst-case performance in deep learning regimes. This suggests an augmentation over the small portion of data points identified by DRO, instead of counting on learning over the small portion of data points only, and leads to a similar design of JTT Liu et al. (2021). However, with the theoretical understanding and method developments, existing approaches dealing with sub-population shifts still suffer risks such as unstable performance across training epochs Zhai et al. (2021). A popular and compromised solution is a domain explicit validation dataset along with an early stopping strategy to guarantee reasonable worst-case performance Zhai et al. (2022), which is also employed in this work.

## 4  L-DRO: DISTRIBUTIONAL ROBUSTNESS VIA LANGUAGE

The ability of the CLIP model that fuses natural language supervision into the training phase for various purposes leaves great untouched potential. A set of text prompts with carefully designed class-domain descriptions can construct a zero-shot classifier that works reasonably well for specific tasks. Inspired by the construction, this work proposes to use natural language information to mitigate sub-population shifts.

*Using language to mitigate sub-population shift* Note that the sub-populations $\{Q_1, ..., Q_n\}$ is defined over space $\mathcal{X} \times \mathcal{Y}$, i.e., for a specific sub-population, e.g., $Q_k$, the sub-population is decided by the combination of attributes from input space and label space together. Take a typical setting on CelebA as an example, attributes from label space $\{\mathbf{y}_b, \mathbf{y}_n\} := \{\text{blond, not blond}\}$ and attributes from input space $\{F, M\} := \{\text{female, male}\}$ consist of four sub-populations $\{\text{blond male, not blond male, blond female, not blond female}\}$ for worst-case performance estimation.

Since label space information is explicit, the performance over label space can be perfectly balanced. Intuitively, if the performance of sub-populations within the same class can be balanced as well, the sub-population shifts can be mitigated. Using the principle of applying ERM for deep model training with the balanced dataset for label space $\mathcal{P}(\mathbf{y}_b|\mathbf{x}) \approx \mathcal{P}(\mathbf{y}_n|\mathbf{x})$, the performance of sub-populations is given by $\mathcal{P}(\mathbf{y}_b, F|\mathbf{x}) : \mathcal{P}(\mathbf{y}_n, F|\mathbf{x}) : \mathcal{P}(\mathbf{y}_b, M|\mathbf{x}) : \mathcal{P}(\mathbf{y}_n, M|\mathbf{x}) \approx \mathcal{P}(F|\mathbf{x}) : \mathcal{P}(F|\mathbf{x}) : \mathcal{P}(M|\mathbf{x}) : \mathcal{P}(M|\mathbf{x})$. This means the performance of a specific sub-population is decided by the proportion of the sub-population over the dataset. Further, $\ell_{\text{ent}}(\mathcal{P}(F|\mathbf{x}), \mathcal{P}(M|\mathbf{x}))$ can serve as a measure to sub-population shifts, formulated as:

$$\sup_{Q \in \mathcal{Q}} \mathbb{E}_{Z \sim Q}[\ell(\theta, Z)] \propto -\ell_{\text{ent}}(\mathcal{P}(F|\mathbf{x}), \mathcal{P}(M|\mathbf{x})). \tag{3}$$

It suggests that sub-population shift can be naturally mitigated through $\max \ell_{\text{ent}}(\mathcal{P}(F|\mathbf{x}), \mathcal{P}(M|\mathbf{x}))$, and achieving the goal does not rely on instance-wise label information.

Based on intuition, we proposed Language-based Distributional Robust Optimization, abbreviated as L-DRO. L-DRO is built upon equation 3 and aims at improving the worst-case performance of CLIP without instance-wise label information.

- Following architecture design in equation 1, we propose using the sup-population descriptions to debias the original feature representations.
- In order to retain general performance, another objective is designed to maintain consistency between the original feature representations and the debiased ones.

Given image encoder of CLIP $I(\cdot)$, text encoder $T(\cdot)$, adapter $A(\cdot)$, *classification text prompt* (target domain descriptions ) $\{t_1, \cdots, t_c\}$, and *debiasing text prompt* (a set of semantically opposite sub-population descriptions) $\{\hat{t}_1, \cdots, \hat{t}_s\}$, the objective of L-DRO is formulated as:

$$\min_{\theta_A} \ell(\mathbf{x}, \{\hat{t}_1, \cdots, \hat{t}_s\}) := \min_{\theta_A} 1 - \ell_{\text{ent}}\Big([((A_{\theta_A} \circ I(\mathbf{x}))^T T(\hat{t}_i)]_{i \in [s]}\Big) - \eta \cdot \ell_{\text{sim}}\big(I(\mathbf{x}), A_{\theta_A} \circ I(\mathbf{x})\big),$$
$$(4)$$

where $\ell_{\text{ent}}(\mathbf{a}) := -\text{softmax}(\mathbf{a})^T \log(\text{softmax}(\mathbf{a}))$ is the entropy loss that encourages the inability to distinguish across sub-populations using the learned feature representation, i.e., debiasing the feature representations. And $\ell_{\text{sim}}(\mathbf{a}, \mathbf{b}) := \frac{\mathbf{a}^T \mathbf{b}}{||\mathbf{a}||||\mathbf{b}||}$ is the consistency loss (cosine similarity) that encourages the similarity of feature representation before and after the adapter. $\eta$ is a scalar to balance the above two terms. The corresponding training and inference of L-DRO follow the procedures of Eqs. equation 1 and equation 2. An outline of the training phase of our method is shown in Figure 1.

## 5 EXPERIMENTS

In this section, we begin by showcasing the consistent improvement of L-DRO over zero-shot learning in terms of worst-case performance. Specifically, we investigate this improvement and highlight the stability of L-DRO across different training epochs (see Section 5.1). Subsequently, we examine the impact of debiasing on various sub-populations, including both independent and correlated sub-populations (see Section 5.2). Further, we explore the potential of the debiased feature representations to stabilize the existing methods that deal with sub-population shifts. The performance metrics include average accuracy and worst-case accuracy, and the worst-case accuracy represents the lowest accuracy observed among the different subpopulations. To ensure the robustness and reliability of the results, each experiment is repeated 10 times using different random seeds to get the mean and standard deviation of accuracy.

**Model architecture** CLIP Radford et al. (2021) is selected as the vision-language foundation model in the experiment, and the training and inference follow equation 1 and equation 2. The subspace mapping $A_{\theta_A}(\cdot)$ is a two-layer MLP with the same input and output dimensions.

**Dataset and pre-defined sub-populations** Most of our experiments were evaluated on CelebA Liu et al. (2015), which is a large-scale face attributes dataset with 40 attribute annotations for each image. The target task of CelebA is to differentiate if a given human face image is blond hair or not blond hair. The attributes from input space for constituting sub-population shifts are selected as {male, female}, then we have four sub-populations {blond male, not blond male, blond female, not blond female} on CelebA dataset for worst-case performance estimation. Another selected and commonly used dataset is Waterbirds (95%) Sagawa et al. (2019), which asks classifiers to differentiate if a given image is waterbirds or landbirds. The training set of Waterbirds places 95% of all waterbirds against a water background with the remaining 5% against a land background. And the similar unbalanced setting also applied to the other class landbirds. Then we have four sub-populations {waterbird in water, waterbird in land, landbird in water, landbird in land } on Waterbirds dataset for worst-case performance estimation.

We note that the best model is selected based on a domain-aware validation dataset across varying hyper-parameters and training epochs (early stopping strategy), please refer to Appendix B.2 of Zhai et al. (2021) and Zhai et al. (2022) for a comprehensive discussion about the necessity of domain-aware validation dataset for model selection in methods dealing with sub-population shifts.

### 5.1 WORST-CASE PERFORMANCE AND BEYOND

**The selection of text prompt** The selection of appropriate text prompts, including classification and debiasing prompts, significantly impacts the performance of CLIP. Thus, our initial investigation focuses on examining the impact of different text prompts on both average accuracy and worst-case

Table 1: Under CelebA and CLIP (ViT-B/32 and RN50), the Average Accuracy (Avg.Acc.) and Worst-Case Accuracy (W.C.Acc.) over sub-populations with varying classification text prompt and debiasing text prompt.(%)

| Classification text prompt And Debiasing text prompt | Method | RN50 (Avg.Acc & W.C.Acc.) | ViT-B/32 (Avg.Acc & W.C.Acc.) |
|---|---|---|---|
| "a photo of []" And "a photo of []" | Zero-shot | 64.9 & 49.2 | 49.6 & 35.6 |
| | L-DRO | 66.3±0.8 & **55.4±1.0** | 51.6±1.2 & **38.3±1.5** |
| "photo of a [] people" And "photo of a [] people" | Zero-shot | 70.1 & 52.9 | 58.8 & 49.2 |
| | L-DRO | 70.9±0.8 & **53.3±1.4** | 60.4±0.7 & **53.3±0.9** |
| "photo of a [] hair people" And "photo of a [] people" | Zero-shot | 77.4 & 65.2 | 86.4 & 61.1 |
| | L-DRO | 78.3±0.8 & **66.5±1.5** | 86.1±0.3 & **69.7±2.1** |
| "a photo of a [] hair people And "a photo of a [] people" | Zero-shot | 77.4 & 62.6 | 85.2 & 70.6 |
| | L-DRO | 78.0±0.7 & **63.5±1.4** | 83.6±0.3 & **79.2±1.3** |
| "picture of [] hair people" And "picture of [] people" | Zero-shot | 87.6 & 67.1 | 86.1 & 65.0 |
| | L-DRO | 88.8±0.3 & **74.7±1.1** | 85.0±0.5 & **65.4±1.3** |
| "a picture of [] hair people" And "a picture of [] people" | Zero-shot | 88.8 & 75.6 | 87.3 & 75.0 |
| | L-DRO | 89.0±0.3 & **75.8±1.1** | 85.2±0.7 & **75.9±0.8** |
| "a picture of [] people" And "a picture of [] people" | Zero-shot | 83.6 & 75.3 | 71.6 & **66.3** |
| | L-DRO | 85.7±0.4 & **78.9±1.7** | 71.2±0.4 & 58.8±0.8 |

 in the classification text prompt is male or female and [] in the classification text prompt is not blond or blond.

accuracy. Table 1 shows that L-DRO consistently outperforms zero-shot learning across most of popular text prompt selections for both network architectures RN50 and ViT-B/32. Meanwhile, the average accuracy increases most time surprisingly since the objective of L-DRO, equation 4, does not involve any label information about the target task. More investigations in text prompts for Waterbirds dataset and other settings are detailed in Appendix A.

**Performance cross datasets and network architectures** Table 2 shows the performance of L-DRO with different network architectures on CelebA and Waterbirds. Compared with other baselines, under both network architectures RN50 and ViT-B/32 with both CelebA dataset and Waterbirds dataset, L-DRO outperforms all of the baselines with great improvement. However, with network architectures ViT-L/14, the L-DRO fails to debias the feature representations and improve the performance compared with zero-shot learning, we investigate the phenomenon in Appendix C, which shows that text prompts on CLIP (ViT-B/32) does not consistently translate to high performance on CLIP (ViT-L/14). Unless otherwise specified, we use ViT-B/32 for the following experiments.

**Stable worst-case accuracy across training epochs** We highlight the performance stability of L-DRO across training epochs since most methods dealing with sub-population shifts suffer the instability of performance along with training epochs. A domain-aware validation dataset is usually necessary to obtain reasonable worst-case performance. As we can see from Figure 2, baseline DRO methods, $\chi^2$-DRO Hashimoto et al. (2018), JTT Liu et al. (2021), and CVaR DRO Namkoong & Duchi (2016), show large fluctuations in both average accuracy and worst-case accuracy throughout the training epochs. On the other hand, Figure 2 shows that the average accuracy and worst-case accuracy of L-DRO under the default setting where $\eta = 0.2$ are significantly stable along with training epochs, which underscore the advancement of methods based on re-weighting or augmentation over dataset that bring instability during training referring Figure 2 in Zhai et al. (2021).

**Data efficiency** Benefiting from the parameter-efficient training procedure, L-DRO demonstrates data efficiency. Table 3 summarizes the average accuracy and worst-case accuracy with varying sizes of training data. The performance gain compared with zero-shot learning starts from 2048 training examples under the CelebA dataset, and the gain for Waterbirds starts from 512 training examples.

## 5.2 MUTUAL EFFECTS OF DEBIASING VARIOUS SUB-POPULATIONS AND BEYOND

We investigate the effect of unaligned debiasing and the combination of multiple debiasing sources at Table 4. The first eight rows in Table 4 show that unaligned debiasing generally has similar

Table 2: The average accuracy and worst-case accuracy of OrthCali Chuang et al. (2023), ERM, CVaR DRO Namkoong & Duchi (2016), $\chi^2$-DRO Hashimoto et al. (2018), JTT Liu et al. (2021), and L-DRO (our proposed method) over different datasets and network architectures.[1] (%)

| Architecture | Method[2] | CelebA | | Waterbirds | |
|---|---|---|---|---|---|
| | | Avg. Acc. | W.C. Acc. | Avg. Acc. | W.C. Acc. |
| RN50 | | | | | |
| $I \triangleright T$ | OrthCali | 52.5 | 24.6 | 69.8 | 60.4 |
| $I \triangleright A \triangleright T$ | ERM | 95.2±0.1 | 41.2±3.6 | 83.0±1.1 | 59.6±2.1 |
| $I \triangleright A \triangleright T$ | CVaR DRO | 87.0±3.4 | 70.6±4.7 | 77.2±3.7 | 59.5±4.2 |
| $I \triangleright A \triangleright T$ | $\chi^2$-DRO | 88.1±3.8 | 64.2±12.8 | 79.0±1.8 | 61.1±3.1 |
| $I \triangleright A \triangleright T$ | JTT | 89.4±2.5 | 49.7±4.7 | 78.5±2.6 | 58.3±6.0 |
| $I \triangleright A \triangleright T$ | L-DRO | 85.7±0.5 | **78.9±1.7** | 77.4±1.3 | **62.7±2.8** |
| ViT-B/32 | | | | | |
| $I \triangleright T$ | OrthCali | 73.1 | 60.8 | 83.9 | 59.7 |
| $I \triangleright A \triangleright T$ | ERM | 95.3±0.1 | 44.2±2.5 | 83.3±1.0 | 59.7±2.4 |
| $I \triangleright A \triangleright T$ | CVaR DRO | 84.8±4.9 | 67.1±10.4 | 78.3±3.2 | 60.5±4.2 |
| $I \triangleright A \triangleright T$ | $\chi^2$-DRO | 87.4±4.5 | 72.0±9.6 | 79.3±2.9 | 59.3±5.7 |
| $I \triangleright A \triangleright T$ | JTT | 90.3±2.1 | 53.4±2.6 | 80.3±1.9 | 60.5±3.0 |
| $I \triangleright A \triangleright T$ | L-DRO | 83.6±0.3 | **79.2±1.3** | 77.6±0.5 | **64.8±0.8** |

[1] The proposed method L-DRO and {CVaR DRO, $\chi^2$-DRO, JTT} requires a domain-aware validation dataset for hyper-parameter selection. OrthCali is training-free regimes and generally don't need a validation dataset except for selecting a better prompt. And, only adapter $A$ is tunable.

[2] CVaR-DRO and $\chi^2$-DRO use the two-phase training strategy same with JTT. The motivation is detailed in section 3.2, and the performance comparison is demonstrated in Appendix B.

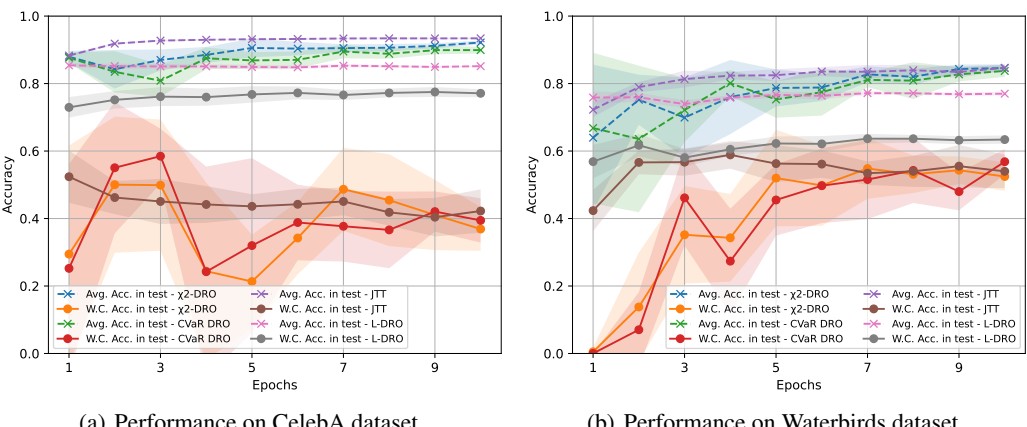

(a) Performance on CelebA dataset.  (b) Performance on Waterbirds dataset.

Figure 2: Under CLIP (ViT-B/32), the average and worst-case accuracy of validation dataset and test dataset over epochs using L-DRO and baseline methods. The left figure shows the performance of CelebA dataset, and the right figure shows the performance of Waterbirds dataset.

performance with zero-shot learning, i.e., basically, the L-DRO works on influential attributes only and will not hurt the embedding of uncorrelated attributes. The last two rows demonstrate the performance of combining multiple debiasing sources, it shows that L-DRO can debias multiply influential attributes at the same time and the performance is slightly degraded compared with independent debiasing. In Table 5, we further investigate using L-DRO to debias semantically correlated source sub-populations and target sub-populations, and it shows that a degressive correlation between source

Table 3: Under CLIP (ViT-B/32), the average accuracy and worst-case accuracy with varying sizes of training data. The accuracy with $\star$ reaches the performance of zero-shot learning, and the accuracy with $\diamond$ reaches or is close to the performance of training with full data.(%)

| Dataset | Acc. | Performance over varying size of training data | | | | | |
|---|---|---|---|---|---|---|---|
| | | 512 | 1024 | 2048 | 4096 | 8192 | 16384 |
| CelebA | Avg. | 77.1±8.5 | 84.6±1.1 | 84.7±0.8 | 84.5±0.8 | 83.9±0.6 | 83.7±0.9 |
| | W.C. | 57.2±6.4 | 69.4±6.2 | 71.4±4.6$\star$ | 74.1±2.8 | 76.3±3.3 | 77.4±2.8$\diamond$ |
| Waterbirds | Avg. | 74.4±1.8 | 76.0±0.7 | 77.8±0.7 | 78.4±0.8 | / | / |
| | W.C. | 57.9±3.1$\star$ | 63.2±1.9 | 65.1±1.1$\diamond$ | 66.5±1.4 | / | / |

Table 4: Under CelebA and CLIP (ViT-B/32), the average accuracy and worst-case accuracy over unaligned source sub-populations and target sub-populations. (%)

| Method | Unaligned debiasing | | Avg. Acc. | W.C. Acc. |
|---|---|---|---|---|
| | Source | Target | | |
| Zero-shot | - | {male, female} | 85.2 | 70.6 |
| L-DRO | {male, female} | {male, female} | 83.6±0.3 | 79.2±1.3 |
| L-DRO | {old, young} | {male, female} | 84.5±0.3 | 69.5±1.3 |
| L-DRO | {attractive, not attractive} | {male, female} | 84.3±0.2 | 73.9±1.7 |
| L-DRO | {straight hair, wavy hair} | {male, female} | 85.8±0.4 | 67.7±2.2 |
| Zero-shot | - | {old, young} | 85.1 | 73.5 |
| L-DRO | {old, young}[1] | {old, young} | 88.0±0.7 | 84.3±1.6 |
| L-DRO | {male, female} | {old, young} | 84.3±0.8 | 69.4±1.7 |
| L-DRO | {attractive, not attractive} | {old, young} | 84.0±0.7 | 79.6±1.7 |
| L-DRO | {straight hair, wavy hair} | {old, young} | 85.9±0.5 | 72.1±1.8 |
| L-DRO | [{old, young}, {male, female}] | {male, female} | 86.7±0.8 | 78.9±1.6 |
| | | {old, young} | 86.9±1.1 | 81.2±1.8 |

[1] Debiasing test prompt for input space attributes {old, young} is "a photo of a [{old, not old}, {young, not young}] people".

Table 5: Under CelebA and CLIP (ViT-B/32), the average accuracy and worst-case accuracy over semantically correlated source sub-populations and target sub-populations. The semantic relations refer University (2010). (%)

| Method | Semantically correlated debiasing | | Avg. Acc. | W.C. Acc. |
|---|---|---|---|---|
| | Source | Target | | |
| Zero-shot | - | {male, female} | 85.2 | 70.6 |
| L-DRO | {male, female} | {male, female} | 83.6±0.3 | 79.2±1.3 |
| L-DRO | {man, woman}[1] | {male, female} | 84.9±0.3 | 75.8±2.3 |
| L-DRO | {boy, girl}[2] | {male, female} | 85.1±0.4 | 74.4±2.5 |
| Zero-shot | - | {old, young} | 85.1 | 73.5 |
| L-DRO | {old, young} | {old, young} | 88.0±0.7 | 84.3±1.6 |
| L-DRO | {adult, juvenile} | {old, young} | 81.8±0.5 | 79.1±0.6 |
| L-DRO | {mature, immature} | {old, young} | 85.6±0.3 | 71.2±1.4 |

[1] Debiasing test prompt is "a photo of a {man, woman}".
[2] Debiasing test prompt is "a photo of a {boy, girl}".

and target sub-populations generally gets decreasing performance on the worst-case accuracy as expected.

Table 6: Under CLIP (ViT-B/32), the average accuracy and worst-case accuracy of ERM, L-DRO+ERM, CVaR DRO Namkoong & Duchi (2016), L-DRO+CVaR-DRO, $\chi^2$-DRO Hashimoto et al. (2018), L-DRO+$\chi^2$-DRO, JTT Liu et al. (2021), and L-DRO + JTT over different datasets. (%)

| Architecture[a] | Method | CelebA | | Waterbirds | |
|---|---|---|---|---|---|
| | | Avg. Acc. | W.C. Acc. | Avg. Acc. | W.C. Acc. |
| $I \triangleright A^2 \triangleright T$ | ERM | 95.3±0.1 | 44.2±2.5 | 83.3±1.0 | 59.7±2.4 |
| $I \triangleright A^1 \triangleright A^2 \triangleright T$ | L-DRO + ERM | 95.1±0.1 | 45.3±4.6 | 79.9±2.0 | 55.6±3.3 |
| $I \triangleright A^2 \triangleright T$ | CVaR DRO | 84.8±4.9 | 67.1±10.4 | 78.3±3.2 | 60.5±4.2 |
| $I \triangleright A^1 \triangleright A^2 \triangleright T$ | L-DRO + CVaR-DRO | 85.9±3.2 | 71.0±8.1 | 78.1±3.0 | 56.7±4.5 |
| $I \triangleright A^2 \triangleright T$ | $\chi^2$-DRO | 87.4±4.5 | 72.0±9.6 | 79.3±2.9 | 59.3±5.7 |
| $I \triangleright A^1 \triangleright A^2 \triangleright T$ | L-DRO + $\chi^2$-DRO | 84.7±2.8 | 73.8±6.8 | 77.5±4.4 | 57.1±5.2 |
| $I \triangleright A^2 \triangleright T$ | JTT | 90.3±2.1 | 53.4±2.6 | 80.3±1.9 | 60.5±3.0 |
| $I \triangleright A^1 \triangleright A^2 \triangleright T$ | L-DRO + JTT | 91.2±2.2 | 48.4±2.8 | 78.2±2.0 | 56.7±2.3 |

[a] $I \triangleright A^1 \triangleright A^2 \triangleright T$ denotes that L-DRO tunes the first adapter $A^1$ which is the same adapter as $A_{\theta_A}(\cdot)$, and the second embedded method tunes the second adapter $A^2$ which is a three-layer MLP with the same input and output dimensions.

### 5.3 DEBIASED FEATURES IMPROVING THE STABILITY OF METHODS DEALING WITH SUB-POPULATION SHIFTS

In this section, we empirically verify that L-DRO has the potential to help stabilize the existing methods including CVaR-DRO Namkoong & Duchi (2016), $\chi^2$-DRO Hashimoto et al. (2018), and JTT Liu et al. (2021) that deal with sub-population shifts based on loss values or wrongly classified instances. In Table 6, we compare the performance of some commonly used DRO methods before and after using the adapter to debias feature on CelebA dataset and Waterbirds dataset. Under CelebA dataset, results show that L-DRO+CVaR-DRO and L-DRO+$\chi^2$-DRO improves the mean and standard deviation of worst-case accuracy over the original CVaR-DRO and $\chi^2$-DRO. But, there is no improvement when using JTT. Under Waterbirds dataset, the combination does not show any advantages. However, it seems that if the base performance has reasonable results, then L-DRO can further improve its performance upon it. If not, such as the first four rows of performance on Waterbirds, the DRO methods cannot achieve better performance than ERM, then the combination fails either.

## 6 CONCLUSION

In this work, we studied the sub-population shift in the multimodality model CLIP, where sub-population shifts in one modality can be described and defined in another modality. Specifically, one can use natural language to describe the influential attributes that cause the shifts, then those descriptions can be mapped into the space same as image embedding. To this end, we proposed L-DRO to debias the image representations according to the vectorized influential attributes descriptions and exploit the debiased representations to achieve better performance while sub-population shifts exist during testing. Compared with zero-shot learning, L-DRO shows improved worst-case performance under domain oblivious settings and occasionally even enhances average performance without instance-wise label information. In L-DRO, we introduced the use of entropy and consistency terms to facilitate the cooperation between the two modalities, focusing specifically on the concerned attributes while minimizing their impact on other factors.

**Limitations.** Our work has several limitations that should be acknowledged. Firstly, as is the case with other studies utilizing vision-language models as the foundation, the selection of text prompts, including classification and debiasing prompts in this work, plays a crucial role in achieving reasonable average accuracy and worst-case accuracy. Therefore, even though we can eliminate the need for a domain-oblivious validation dataset during training, careful consideration and appropriate selection of prompts are necessary and deserve a comprehensive study in our future work. Additionally, we anticipate that there may be certain sub-populations that prove challenging to describe accurately, which could hinder the application of our proposed method.

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

# A    THE EFFECT OF TEXT PROMPT

Table 7: Under CelebA and CLIP (ViT-B/32), the average accuracy and worst-case accuracy over sub-populations with varying classification text prompt and debiasing text prompt. (%)

| Classification text prompt And Debiasing text prompt | Method | Avg. Acc. | Worst-case Acc. |
|---|---|---|---|
| Input space sub-group: {female, male} | | | |
| "a photo of a {not blond, blond} hair people" | Zero-shot | 85.2 | 70.6 |
| And "a photo of a {female, male} people"[1] | L-DRO | 83.6±0.3 | **79.2±1.3** |
| Or "a photo of a {female, not female} people" | L-DRO | 88.8±0.3 | 65.0±0.9 |
| Or "a photo of a {male, not male} people" | L-DRO | 89.4±0.3 | 37.8±2.2 |
| Or "a photo of a {[female, not female], [male, not male]} people" | L-DRO | 89.9±0.3 | 60.7±2.4 |
| Input space sub-group: {old, young} | | | |
| "a photo of a {not blond, blond} hair people" | Zero-shot | 85.1 | 73.5 |
| And "a photo of a {old, young} people" | L-DRO | 84.4±0.3 | 74.5±1.5 |
| Or "a photo of a {old, not old} people" | L-DRO | 82.2±0.05 | 78.2±0.6 |
| Or "a photo of a {young, not young} people" | L-DRO | 91.3±0.1 | 51.6±1.9 |
| Or "a photo of a {[old, not old], [young, not young]} people" | L-DRO | 88.0±0.7 | **84.3±1.6** |

[1] " " denotes default choice.

Table 8: Under Waterbirds and CLIP (ViT-B/32 and RN50), the Average Accuracy (Avg.Acc.) and Worst-Case Accuracy (W.C.Acc.) over sub-populations with varying classification text prompt and debiasing text prompt.(%)

| Classification text prompt And Debiasing text prompt | Method | RN50 (Avg.Acc & W.C.Acc.) | ViT-B/32 (Avg.Acc & W.C.Acc.) |
|---|---|---|---|
| "a {landbird, waterbird}" And "{water, land}" Or "{water, forest}" | Zero-shot L-DRO L-DRO | 68.1 & 43.4 72.6±1.2 & 49.5±2.7 74.9±1.2 & **57.6±2.6** | 74.8 & 56.8 75.1±1.6 & 56.6±2.6 77.6±0.5 & **64.8±0.8** |
| "photo of {landbird, waterbird}" And "photo of {water, land}" | Zero-shot L-DRO | 66.3 & **43.2** 63.3±1.4 & 41.0±3.3 | 66.1 & 39.6 76.0±0.7 & **61.9±1.4** |
| "photo of a {landbird, waterbird}" And "photo of a bird on {water, land}" | Zero-shot L-DRO | 78.1 & 34.0 74.3±0.9 & **57.9±1.8** | 68.7 & 43.6 71.8±2.5 & **49.7±4.7** |
| "photo of a {landbird, waterbird}" And "photo of a bird on {water, land} background" | Zero-shot L-DRO | 78.1& 34.0 77.4±1.3 & **62.7±2.8** | 68.7 & 43.6 70.0±3.2 & **46.9±4.8** |
| "a photo of a {landbird, waterbird}" And "a photo of a bird on {water, land}" | Zero-shot L-DRO | 76.8 & 40.8 73.9±3.0 & **54.4±4.6** | 69.7 & 45.5 71.4±3.4 & **50.2±5.2** |
| "a photo of a {landbird, waterbird}" And "a photo of a bird on {water, land} background" | Zero-shot L-DRO | 76.8 & 40.8 75.3±0.8& **58.1±1.7** | 69.7& **45.5** 67.5±2.9 & 43.9±4.2 |

## B EFFECTS OF TWO-PHASE TRAINING ON DRO METHODS

Table 9: The average accuracy and worst-case accuracy over different datasets and methods.[1] (%)

| Dataset | Architecture | Method | Average Acc. | Worst-case Acc. |
|---------|--------------|--------|--------------|-----------------|
| CelebA | $I \triangleright A^2 \triangleright T$ | ERM | 95.3±0.1 | 44.2±2.5 |
| | $I \triangleright A^2 \triangleright T$ | CVaR DRO | 86.6±1.0 | 11.7±9.7 |
| | $I \triangleright A^2 \triangleright T$ | $\chi^2$-DRO | 84.2±8.3 | 61.3±8.5 |
| | $I \triangleright A^2 \triangleright T$ | CVaR DRO$^\star$ | 84.8±4.9 | 67.1±10.4 |
| | $I \triangleright A^2 \triangleright T$ | $\chi^2$-DRO$^\star$ | 87.4±4.5 | 72.0±9.6 |

[1] Keeping the same settings with Table 6. And $^\star$ denotes using the same two-phase training strategy with JTT, and the method without $^\star$ denotes the original version (mini-batch) of CVaR DRO and $\chi^2$-DRO.

## C TEXT PROMPT FOR CLIP (VIT-L/14)

Table 10 reveals that the effectiveness of text prompts on CLIP (ViT-B/32) does not consistently translate to high performance on CLIP (ViT-L/14). Employing "a photo of a { } people" as the prompt for CLIP (ViT-L/14) achieves a more reasonable performance, and the introduction of L-DRO further enhances the overall performance in this context.

Table 10: Under CelebA and CLIP (ViT-L/14), the average accuracy and worst-case accuracy over sub-populations with varying classification text prompt and debiasing text prompt [1].(%)

| Classification text prompt And Debiasing text prompt | Method | Average Acc. | Worst-case Acc. |
|---|---|---|---|
| "a photo of {not blond, blond}" | Zero-shot | 39.1 | 28.8 |
| "photo of a {not blond, blond}" | Zero-shot | 75.9 | 65.2 |
| "a photo of a {not blond, blond}" | Zero-shot | 64.0 | 39.7 |
| "photo of a {not blond, blond} people" | Zero-shot | 80.7 | 77.9 |
| "a photo of a {not blond, blond} people" | Zero-shot | 85.4 | 76.1 |
| "photo of a {not blond, blond} hair people" | Zero-shot | 78.5 | 70.7 |
| "a photo of a {not blond, blond} hair people" | Zero-shot | 75.6 | 64.5 |
| And "a photo of a {male, female} people"[2] | L-DRO | 85.9±0.9 | **79.7±1.9** |

[1] classification and debiasing text prompts use the same structure, e.g., "a photo of a { } people" will be used for both classification and debiasing text prompts.
[2] " " denotes default choice.

