# OpenReview forum: "Cross-modality debiasing: using language to mitigate sub-population shifts in imaging"
_ICLR.cc/2024/Conference — Submitted to ICLR 2024_

### Official Review · Reviewer_KXPh · 2023-11-01

**Soundness:** 3 good
**Presentation:** 3 good
**Contribution:** 2 fair
**Rating:** 5
**Confidence:** 4

**Summary:**

*Note that this manuscript was previously submitted to NeurIPS 2023, where it was withdrawn following scores that would have likely led to rejection. I was assigned a reviewer for this paper at the time as well, and from what I can tell the paper remains entirely unchanged. Hence, I am going to take the liberty of resubmitting my previous review entirely unchanged as well.*

This paper addresses the problem of subgroup robustness, i.e., the concern that a model shows markedly worse performance for specific subpopulations. Note that a subpopulation here is defined across both label and input space. Mathematically this translates to $\min_\theta \sup_{Q\in\mathcal{Q}}\mathbb{E}_{Z\sim Q}[\ell(\theta,Z)]$ where $\mathcal{Q}$ is a set of subpopulations and $\ell$ is the loss for a given sample, $Z$, and parameters $\theta$. That is, the goal is to maximize the worst-case performance of the classifier over all subpopulations.

The suggested approach in this paper relies on a vision-language model: The language model is used to describe the subpopulations in input space. For example, if subpopulations are defined as blond/not blond (label space) men/women (input space), then "a photo of a man/woman" ("debiasing prompt") is fed and used to generate embeddings $T(\hat{t}_i)$ (see equation 4). An adapter $A$ is then trained to "debias" the vision embeddings, which means that the embeddings lie equally far from all $T(\hat{y}_i)$ (e.g., it is no longer possible to tell the gender from the image features). At the same time, a consistency loss using the un-adapted features ensures that embeddings don't collapse.

The idea here is that if the classes are balanced (e.g., there are as many blond as not blond people in the dataset) and the vision features are debiased (e.g., there is no more information in there about gender) then the performance of the classifier must end up being equal across all of the (four) subpopulations (blond/not blond men/women).

Experiments show significant improvements in the worst-case accuracy for all models except ViT-L/14 on the CelebA dataset (subpopulations are blond/not blond men/women) and the Waterbirds dataset (subpopulations are waterfowl/landfowl in the air/on land). Experiments show that several hundred/thousand examples are needed to see improvements over the baseline (i.e., no debiasing).

A few other experiments show that the method can be used in conjunction with other methods (e.g., CVaR-DRO and $\chi^2$-DRO) and that it is possible to debias multiple attributes at the same time.

**Strengths:**

The proposed idea is appealing in its simplicity: A vision-language model can be used to describe which attributes should be protected/robust (e.g., male/female) and an adapter is quite directly trained to remove these features entirely from the image features.

The experimental results are encouraging. Improvements seem reliant on prompt engineering and not all improvements are equally large, but they seem consistent.

**Weaknesses:**

The main strength of the method also seems like a weakness: It relies on a multi-modal model to provide the zero-shot learning capability needed to identify subpopulations in the label space. This makes the method pretty specific.

A concern regarding the applicability of this method is that the experiments show that this approach requires significant prompt engineering. But the assumption in this paper seems to be that there are no labels available (e.g., no male/female labels). That means that it wouldn't be possible to compare the performance of prompts like it was done, e.g., in table 1, right?

All in all, I don't think this is a bad paper. The idea is clean and simple, and the results show that it works. However, I'm wondering if the idea is fleshed out enough, and it leaves me with several practical questions: How does one select a debiasing prompt when there is no ground truth labels? Can I use multiple debiasing prompts? How do I know if I have enough data? I am not sure if all these questions should be left to future work (I think these are more important than questions about how L-DRO interacts with other methods, for example). This is why I am recommending a weak reject.

**Questions:**

Some questions:

* Did the authors explore why their model fails on ViT-L/14? This seems like useful information for practitioners. For example, can they only expect this model to work on smaller models?
* My understanding is that the results in table 6 should be comparable to the 3rd and 7th rows in table 2? There zero-shot learning gets 70.6% worst-case performance. It seems that $\chi^2$-DRO is the only baseline that gets higher than this (but with huge variance)? It seems odd that all the DRO methods do worse than not doing anything at all?
* Assuming that it's not possible for a practitioner to select the best prompt based on validation results (since no ground truth labels are available for the subpopulations) it would be useful to know if the method can be extended to support multiple debiasing texts. That is, rather than having to select a single debiasing prompt and hope it is correct, the user could just give many different ones, increasing the chances of having good results (since the spread is quite large, as is evident in table 1).
* I'm surprised at the results in table 3, which seem to suggest that if the dataset isn't large enough, the proposed method actually harms the results. I would find it useful if the authors could (1) provide insight into why this is the case, and (2) provide some guidance on how a practitioner is supposed to know whether or not they have enough data available for this method to apply.

Minor things:

* It seems odd that all the prompts selected are grammatically incorrect. How does the model perform with correct phrases such as "a photo of a blond/non-blond person"/"person who is blond/not blond" and "a photo of a man/woman"/"male/female person"?

---

> ### Author Response · Authors · 2023-11-20
>
> Weakness and Question 3
>
> We agree with the reviewer that  there are challenges in selecting appropriate text prompts when dealing with language (text) related deep models,  and varying text prompts have a significant influence on the final performance, and this work does need a domain explicit validation dataset to guarantee reasonable worst-case performance, as mentioned in Section 3.2 and Section 5 (Experiment) and employed by related work in this line of research. (JTT and $\chi^{2}$-DRO generally use this validation dataset for another purpose, detailed in ‘stable worst-case accuracy across training epochs’ paragraph of section 5.1).
> However, our solution enables fine-tuning an extra adapter (in the last layers) with language guidance to mitigate subpopulations in the label space, which follows the trend of the broader and broader application scenarios of the (large) multi-modal models. We believe this type of condition closely satisfies real-world conditions instead of assuming it is a specific condition.
>
>
> Question 1
>
> Please refer to Appendix C, we have explored the failure mode on ViT-L/14. The root cause is the inconsistency of the effectiveness of text prompts over various models. Employing a prompt that is different from the default one achieves a more reasonable performance, it shows that the proposed method outperforms other baselines, as shown in Table 10.
>
>
>
> Question 2
>
> Table 6 empirically verifies the potential of L-DRO to help stabilize the existing methods since they are orthogonal, and it demonstrates that the proposed method embedded with other baselines generally improves the performance with CelebA but worsens it with the Waterbirds dataset. And, it is not surprising that some baseline methods actually worsen the performance compared with zero-shot performance such as ERM and JTT. Actually, it is well-established that inappropriate fine-tunings damage the distributional robustness of pre-trained models [1,2,3],  which is also mentioned in the introduction.
>
> It should be noted that parameter-efficient tuning, such as tuning the final layers only, demonstrates popularity under certain large model conditions. The existing work of DRO such as JTT typically tunes the whole model.
>
>
> Question 4
>
> It is true that training with fewer data points or data that are not enough to represent the whole distribution manifest performance decreasing. When data is not enough to represent the whole distribution, then the updating direction is hard to find the right one or even in the opposite direction.
> Table 3 shows the proposed method is data efficient and is possibly due to its parameter-efficient structure.
>
>
>
> [1] Mitchell Wortsman, Gabriel Ilharco, Jong Wook Kim, Mike Li, Simon Kornblith, Rebecca Roelofs, Raphael Gontijo Lopes, Hannaneh Hajishirzi, Ali Farhadi, Hongseok Namkoong, et al. Robust fine-tuning of zero-shot models. In Proceedings of the IEEE/CVF Conference on Computer Vision and Pattern Recognition, pp. 7959–7971, 2022.
>
> [2] Ananya Kumar, Aditi Raghunathan, Robbie Jones, Tengyu Ma, and Percy Liang. Fine-tuning can distort pretrained features and underperform out-of-distribution. arXiv preprint arXiv:2202.10054, 2022.
>
> [3] Xin Zhang, Yusuke Iwasawa, Yutaka Matsuo, and Shixiang Shane Gu. Amortized prompt: Lightweight fine-tuning for clip in domain generalization. arXiv preprint arXiv:2111.12853, 2021.

---

> > ### Author Response · Authors · 2023-11-22
> >
> > Dear Reviewer KXPh,
> >
> > We hope our response has answered your questions and clarified the concerns. If there are any further questions, please feel free to inform us before the closure of the interactive rebuttal system.
> >
> > Appreciate your time and happy Thanksgiving.

---

### Official Review · Reviewer_BEYs · 2023-11-05

**Soundness:** 2 fair
**Presentation:** 3 good
**Contribution:** 3 good
**Rating:** 5
**Confidence:** 3

**Summary:**

This paper proposed to utilize the language description of opposite semantics to mitigate the sub-population shifts via CLIP model. This method doesn't require instance-wise label information but instead encourages CLIP's inability to distinguish across sub-populations using the learned features by maximizing the cross-entropy of classifying images into sub-population semantics.  Experimental result demonstrates performance improvement with the proposed method.

**Strengths:**

1. Proposed L-DRO doesn't require instance-level labels. It maximizes the cross-entropy loss to discourage image features too close to any biased descriptions. IMO, it's an effective and novel method.
2. The experiments are very detailed. It studies whether L-DRO can help original CLIP zero-shot inference, how L-DRO is compared with other methods, and whether the L-DRO can help other methods, etc. And the performance of L-DRO is impressive.

**Weaknesses:**

1. I think the consistency loss might be a bit contradictory to the debiasing loss since the consistency loss encourages the adapter's output to be similar to original image features while the debiasing loss wants the adapter's output to be different from original image features. How do the authors think of / solve the problem? Is there any ablation on the coefficient of the consistency loss?

2. It's interesting that L-DRO can also improve average loss in Tab 1. Can authors further explain on this phenomenon?

**Questions:**

1. Shouldn't the `y_p` in `{y_p, y_n} := {blond, not blond}` in the second paragraph of Sec4 be `y_b` to be coherent with the notation in the next paragraph?

================\
After reading other reviews and the authors' responses, I'd like to lower my rating a bit to 5.

---

> ### Author Response · Authors · 2023-11-20
>
> We greatly appreciate your constructive comments. Please see our responses below;
>
> Weakness 1
>
> In our loss design, $\eta$ is expected to balance the two items, consistency loss and similarity loss. And we reported the performance under CelebA dataset and network structure CLIP (ViT-B/32) with varying $\eta$ here to show that the proposed method is not sensitive to the hyper-parameter.
>
> η = 0.1, Avg. Acc. = 0.845±0.006, W.C. Acc. = 0.782±0.016.
>
> η = 0.2 (default), Avg. Acc. = 0.836±0.003, W.C. Acc. = 0.792±0.013.
>
> η = 0.3, Avg. Acc. = 0.845±0.003, W.C. Acc. = 0.771±0.028.
>
> η = 0.4, Avg. Acc. = 0.844±0.004, W.C. Acc. = 0.777±0.020.
>
> η = 0.6, Avg. Acc. = 0.844±0.008, W.C. Acc. = 0.763±0.035.
>
> η = 0.8, Avg. Acc. = 0.842±0.006, W.C. Acc. = 0.771±0.037.
>
> η = 1.0, Avg. Acc. = 0.842±0.007, W.C. Acc. = 0.764±0.040.
>
>
> Weakness 2
>
> We appreciate your careful observation of Table 1 that the improvements in worst-case accuracy seem to slightly improve the average accuracy under varying text prompts with RN50.
> We believe it may imply that our method improves the performance of the worst-case group meanwhile it will least damage the the performance of other groups in such conditions. So that the overall performance (average accuracy) gets improved.
> We conducted the following simple experiments. Under CelebA, CLIP (RN50), and text prompts "a picture of [] people", we report the average accuracy, worst-case accuracy, and additional accuracy for each subgroup over ten seeds.
>
> Zero-shot - acc_avg= 0.837, acc_worst = 0.757, acc_each_group = [0.833, 0.757, 0.808, 0.907],
>
> L-DRO - acc_avg = 0.859, acc_worst = 0.789, acc_each_group = [0.789, 0.814, 0.808, 0.908].
>
> The above results support our claims.
>
>
>
> Question 1
>
> We really appreciate your careful reading and thank you for pointing out the typo (y_p); it has been corrected (See the uploaded revised PDF). Your attention to detail is invaluable.

---

### Official Review · Reviewer_cXZe · 2023-11-09

**Soundness:** 3 good
**Presentation:** 4 excellent
**Contribution:** 3 good
**Rating:** 6
**Confidence:** 3

**Summary:**

This paper  proposes a method to mitigate subpopulation shifts within one modality (e.g., vision) by leveraging the robustness inherent in another modality (e.g., text). This is achieved by learning a vision feature adaptor, which is trained by minimizing equation (4) with debiasing prompts. This will encourage the inability to distinguish across sub-population while maintaining a representation space consistent with the learned space. Utilizing the debiased vision representation alongside the original task-relevant classification prompt enhances worst-case accuracy and also yields benefits for average accuracy.

**Strengths:**

1. The paper proposes an effective framework for debiasing subgroup information using aligned representations from Vision Transformers (ViT), which improves the subgroup robustness. This work could inspire future research in this domain.
2. The evaluation is comprehensive and systematic, showcasing effectiveness across multiple datasets and metrics, and providing a comparison with several state-of-the-art baselines. The Table 4 analysis of (dis)alignment within subgroups between the source and target is both innovative and inspiring.
3. The paper is well-organized and clearly presented.

**Weaknesses:**

1. I’m concerned about the innovation w.r.t. proposing a new strategy for subgroup robustness. The concept of gaining robustness from language modality to enhance another is not novel [1], and the idea of learning debiased representations—where domain predictability is removed and task predictability is emphasized—is fairly common.
2.  While the method is motivated by the general notion that aligned representations from multimodal models can share robustness, the algorithm is restrictive — It primarily facilitates the use of text prompts to learn the debiased representation adapter for improved classification. Extending the algorithm to incorporate other combinations of modalities may be challenging, particularly for modalities where designing debiasing prompts (e.g., in vision) might be difficult.

**Questions:**

1. Can L-DRO be adapted for combinations of modalities other than vision and language, or does the methodology intrinsically require assistance from the language modality?
2. Table 4 offers interesting insights. In the rows where the source and target are misaligned, the average accuracy is comparable to that of a zero-shot scenario, however, there is a significant divergence in worst-case accuracy. Does this suggest other subgroups are either adversely affected or disproportionately benefited (so that the average scores can remain similar)? Does this imply that the algorithm still works with other influential attributes even if they are not directly related to the target domain?

---

> ### Author Response · Authors · 2023-11-20
>
> Thank you for taking the time to read and review our paper. We sincerely appreciate your keen interest in the technical aspects of our work and positive feedback. Please see our responses below;
>
> Weakness
>
> We admit the challenges in designing debiasing guidance from other modalities instead of language modality. And, our solution enables fine-tuning an extra adapter in the last layers with language guidance to mitigate subpopulations in the label space, it follows the trend of the broader and broader application scenarios of the (large) multi-modal models, relying on the guidance from natural language in learning (such as in-context learning), and parameter-efficient fine-tuning regime, which we believe closely matches real-world conditions.
>
>
> Question 1 and 2
>
> In current work, there are no such attempts to adapt L-DRO for modalities other than vision and language, however, we believe the principle developed in this work can be easily adapted to other types of data modalities. For Table 4, there is no clear pattern to describe divergences in worst-case accuracy with unaligned semantic relations currently. We do have similar observations that subgroups are (minorly) adversely affected, it serves as fundamentals of combining multiple debiasing sources, specifically, the last two rows.

---

### Official Review · Reviewer_FPNj · 2023-11-10

**Soundness:** 2 fair
**Presentation:** 3 good
**Contribution:** 2 fair
**Rating:** 3
**Confidence:** 4

**Summary:**

This paper addressed the sub-population shift problem, denoting a domain shift within specific sub-groups between training and testing, by proposing distributional robustness via language (L-DRO). To employ a CLIP model as a debiased zero-shot classifier, L-DRO incorporates an extra feature adapter for image embeddings, enhancing the entropy of their predictions on spurious attributions. In experiments, the author demonstrates that L-DRO attains the highest worst-case accuracy on the Waterbirds and CelebA dataset, surpassing $\mathcal{X}^2-$DRO, CVaR DRO, and JTT.

**Strengths:**

- L-DRO exhibits efficient applicability even in scenarios with multiple categories of spurious attributes.
- Unlike other methods, the training procedure of L-DRO maintains stability across epochs, while competitors experience performance fluctuations.

**Weaknesses:**

- One significant concern revolves around the limitation of entropy maximization.
  - As entropy reaches its maximum on a uniform distribution, the analytic solution of $max \ \ell_{ent}$ can be obtained through $(A_{\theta_A} \odot I(\mathbf{x}))^T (T(\hat{t}_1) - T(\hat{t}_2)) = 0$.
  - This occurs when $(A_{\theta_A} \odot I(\mathbf{x})) $ is projected onto the null-space of $(T(\hat{t}_1) - T(\hat{t}_2))$. However, reducing a single rank is insufficient to eliminate the entire spurious representation.
  - The empirical evidence of this limitation can be observed in Table 5 that penalizing semantically correlated sources even increases worst-case accuracy in zero-shot classification.
- Another concern is the relatively modest performance of the proposed method compared to recently published methods. For instance, [1] achieved 92.9% and 88.3% worst-group accuracy on Waterbirds and CelebA, respectively, which is 10% and 30% higher than L-DRO. This underperformance may be attributed to the weak regularization of the entropy regularization
- Baseline implementation
  - The worst-group accuracy of JTT reported in [1] stands at 86.7% and 81.1% on Waterbirds and CelebA, significantly higher than the results presented in this paper.

[1] https://arxiv.org/pdf/2204.02937.pdf

**Questions:**

- Ablation study) Are there any experimental results for the model trained on the loss defined in Eq.(1)?
- Table 3) Could you compare this experiment with the other methods? The table does not convey how L-DRO maintains stability with varying data sizes.
- Table 4) In some rows, the same source was utilized as the target, resulting in the highest worst-case accuracy. Could you please elaborate on the intention behind this choice and the corresponding performance gain?


- Minor corrections)
  - Table 1) Would you consider changing one of the parentheses { } in the prompt to [ ] or ( ) so that readers can distinguish?
  - Eq. and equation are redundantly appeared.

---

> ### Author Response · Authors · 2023-11-20
>
> We greatly appreciate your constructive comments. Please see our responses below;
>
> Weakness 1
>
> We appreciate the insights regarding our method. Projecting all the debiased feature representation onto the null-space of $[T(\hat{t_{1}}), T(\hat{t_{2}})]^{T}$ is not the only goal since our solution also encourages the similarity of feature representations before and after the adapter with a consistency loss.
> Without the consistency loss, we believe all the feature representations may collapse to, e.g., one fixed direction that is perpendicular with $T(\hat{t_{1}})$ and $T(\hat{t_{2}})$.
> However, with our consistency loss and  $\eta$ (scalar to balance the above two terms), our results empirically negate the collapse where the average accuracy regarding the task basically does not change compared with zero-shot learning.
>
> Weakness 2
>
> The works cited ([1]) differ significantly from ours in scope. For work targeting mitigating sub-population shift, there are two categories: a) domain-oblivious in which the membership of each input sample is unknown during training, and b)  domain-aware settings [1].
> Our work targets the setting, domain-oblivious, as well as the selected baselines. However, the mentioned work [1] “re-trains the last linear layer on a small dataset where the backgrounds are not spuriously correlated with the foreground”, which belongs to the second setting, domain-aware.
> The domain-oblivious, as studied in this paper, is a more challenging setting than domain-aware ones such as [1], since the parameter updating does not have any guidance from the “good data”. As mentioned in Section 3.2, popular strategies along this line of work usually find oblivious evidence in the loss signal, and the high-level idea can be formulated as distributional robust optimization.
>
> Weakness 3
>
> Apologies for any confusion. A quick answer to the misalignment is due to this work only tunes a simple adapter network instead of full model tuning such as JTT and other baselines in the original papers. We mentioned this part in our objective Equation 4 and highlighted the architecture as I▷A▷T in Table 2. To further clarify, we added additional footnotes in Table 2 - only adapter A is tunable. (See the uploaded revised PDF).
> Note that existing works of DRO such as JTT typically tune the whole model while in this work and some large model conditions, parameter-efficient tuning, such as tuning the last two layers only, exhibits popularity.
>
>
>
> Question 1
>
> We want to clarify that ERM (Empirical Risk Minimization) in Table 2 is exactly the result you look for. As expected, ERM generally does not have satisfying performance over sub-population shifts.
>
>
> Question 3
>
> In Table 4, this work investigates the effect of unaligned debiasing and the combination of multiple debiasing sources. Firstly, the unaligned debiasing does not improve the performance under sub-population shifts. And, the last two rows in Table 4 demonstrate the performance of combining multiple debiasing sources.
>
>
> Thank you for pointing out the minor issues, and we have  corrected them.
>
> [1] Zhai R, Dan C, Kolter Z, et al. Doro: Distributional and outlier robust optimization[C]//International Conference on Machine Learning. PMLR, 2021: 12345-12355.

---

> ### Author Response · Authors · 2023-11-22
>
> Dear Reviewer FPNj,
>
> We hope our response has answered your questions and clarified the concerns. If there are any further questions that require our attention, please feel free to inform us before the closure of the interactive rebuttal system.
>
> Appreciate your time and happy Thanksgiving.

---

### Meta-Review · Area_Chair_mW8b · 2023-12-11

**Metareview:**

The authors propose a debiasing method for CLIP using language modality, but most of the reviewers agree that the work is not sufficient for a publication yet. Particularly, justification of the entropy maximization and some experimental settings should be more addressed to get the work published.

**Justification For Why Not Higher Score:**

Some of the points mentioned by the reviewers were not sufficiently addressed in the rebuttal.

**Justification For Why Not Lower Score:**

N/A

---

### Decision · Program_Chairs · 2024-01-16

Reject